# Developmental Coordination Disorder and Unhealthy Weight Status in Taiwanese Children: The Roles of Sex and Age

**DOI:** 10.3390/children10071171

**Published:** 2023-07-06

**Authors:** Yao-Chuen Li, Yu-Ting Tseng, Fang-Yu Hsu, Hsuan-Chu Chao, Sheng K. Wu

**Affiliations:** 1Department of Physical Therapy, China Medical University, Taichung 406040, Taiwan; yaochuenli@mail.cmu.edu.tw (Y.-C.L.); tammyhfy@gmail.com (F.-Y.H.); jay0127680118@gmail.com (H.-C.C.); 2Department of Kinesiology, National Tsing-Hua University, Hsinchu 300013, Taiwan; yutingtseng@mx.nthu.edu.tw; 3Research Center for Education and Mind Sciences, National Tsing-Hua University, Hsinchu 300013, Taiwan; 4Department of Sport Performance, National Taiwan University of Sport, Taichung 404401, Taiwan

**Keywords:** developmental coordination disorder, prevalence, underweight, overweight, obesity

## Abstract

This study aimed to provide up-to-date information regarding the estimated prevalence of developmental coordination disorder (DCD) in Taiwanese children. Their unhealthy weight status was also examined, as well as the roles of sex and age. This cross-sectional study recruited 825 children aged between 4 and 12 years and categorized them into either typically developing children (TD, >16th percentile) or children at risk for DCD (rDCD, ≤16th percentile) based on the result of the Movement Assessment Battery for Children—2nd edition. Body mass index was calculated to determine weight status (i.e., normal weight, underweight, overweight, and obesity). The estimated prevalence of rDCD was 9.7%. There were significantly more boys in the rDCD groups (*p* < 0.001). Additionally, preschool children with rDCD were at greater risk for being underweight. However, there was no significant group difference which was found for overweight/obesity. A lower prevalence of rDCD was found in this study. Nevertheless, children with rDCD may be more likely to be underweight in early childhood. Early intervention should be provided to target this population.

## 1. Introduction

Unhealthy weight, including underweight, overweight, or obesity, has been reported to be a global child health issue by the World Health Organization as these non-communicate disorders might be strongly associated with a variety of physical or mental health problems, such as coronary heart disease, high blood pressure, internalizing or externalizing problems [1,2,3]. In Taiwan, the prevalence of childhood underweight and overweight/obesity were respectively estimated to be 5.6% and 26.8% in preschool children [4], whereas they were 8.9% and 25.4% in primary school students, respectively [5]. This prevalence is concerning as it is higher than in many other Eastern or Western countries [6], indicating that Taiwanese children may be at increased risk for unhealthy weight status.

The issue of childhood overweight and obesity is particularly critical in children with developmental coordination disorder (DCD) [7], which is one of the common neurodevelopment disorders in children and may affect 2–6% of school-aged children [8]. As children with DCD show substantially poor motor coordination, this may hinder them from participating in free play or organized activities which in turn impact their healthy weight status [9,10]. A systematic review has synthesized evidence from 21 eligible studies and concluded that, although the findings may differ by age or sex, school-aged children with DCD would have a higher level of BMI or body fat; furthermore, the overall prevalence of overweight/obesity could even double in this uncoordinated childhood population, compared to the typically developing (TD) peers [7]. This is also the case in Taiwanese children [11,12]. According to a previous study that enrolled 2029 school-aged children with and without DCD, there were more boys and girls with obesity (boys: 33.3%, girls: 38.2%) who had DCD, compared to children with normal weight (boys: 21.5%, girls: 30.5%) [11]. Specifically, children with DCD who showed significant balance problems seemed to be at greater risk for obesity (15.2%) than those without balance problems (7.5–9.1%) [12].

It is worth noting that as underweight has drawn less attention in previous studies that investigated childhood unhealthy weight status in children with DCD, the difference in the prevalence of overweight or obesity may be over-estimated between children with and without DCD and the potential association between poor motor coordination and underweight may be disregarded. Prior research has shown that underweight children could experience more pervasive gross or fine motor problems in their early years [13]. Specifically, due to poor bi-manual dexterity while using utensils, e.g., knives, forks, or chopsticks [8], children with DCD may have difficulties in preparing or eating their meals which may indirectly lead to being underweight. However, there is a lack of existing evidence regarding underweight children with DCD at all ages.

The investigation of unhealthy weight status in Taiwanese children with DCD has not been updated for years, specifically underweight status. Collectively, this study was to conduct a systematic investigation of unhealthy weight status in Taiwanese children and to provide up-to-date evidence concerning the estimated prevalence of DCD in preschool and school-aged populations. Additionally, taking into account the potential impacts of sex and age on weight status [11,14,15], this study also examined the potential roles of sex and age in unhealthy weight in children with DCD. Based on the existing evidence, it was hypothesized that children with DCD would be at greater risk for overweight or obesity and that the prevalence may be higher in girls and increase with age. Nevertheless, due to scarce evidence regarding underweight in children with DCD, this study did not have a proper hypothesis for this specific unhealthy weight status (i.e., underweight).

## 2. Materials and Methods

### 2.1. Participants

This was a cross-sectional study. Children were recruited from two research sites. While 312 children (161 boys, 51.6%) aged between four and six years were from Taichung City (Site I), 522 children (276 boys, 52.9%) aged between seven and twelve years were from Hsinchu City (Site II). Preschool children were reported to have no neurological and musculoskeletal disorders which may impact motor coordination by their parents, whereas the medical condition in school-aged children was confirmed by their teachers. All children did not show intellectual impairment (i.e., IQ < 70). Informed consent was obtained from parents or from both parents and children who were older than seven years old. This study was approved by the Institutional Research Board of China Medical University Hospital (CRREC-108-021).

### 2.2. Procedures

The assessment of motor coordination was administered by trained research assistants using the Movement Assessment Battery for Children—2nd edition (MABC-2). After excluding those who were unable to complete the test (*n* = 6, 0.7%), all children were categorized into two groups based on the testing results: TD (*n* = 748, 90.3%) and at risk for DCD (rDCD, *n* = 80, 9.7%). Children’s body height and weight were then measured and recorded to calculate body mass index (BMI). Data on body height and weight were missed in three children (3/828, 0.4%). Therefore, 825 out of 834 children (98.9%) were included in the final analysis, including 310 (37.6%) preschool children and 515 school-aged children (62.4%).

### 2.3. Measures

#### 2.3.1. Unhealthy Weight Status

In this study, children’s body height and weight were measured to calculate their BMI. Body height was measured using a stadiometer and recorded to the nearest 0.1 cm, whereas body weight was measured using an electronic weight scale and recorded to the nearest 0.1 kg. This study used culturally appropriate criteria with age- and sex-matched norms to determine underweight (<5th percentile), normal weight (5th to <85th percentile), overweight (≧85th percentile), and obesity (≧95th percentile) [16]. Furthermore, unhealthy weight status was defined as underweight, overweight, and obese in this study. Overweight and obese were further merged into the same category for analysis.

#### 2.3.2. Developmental Coordination Disorder

The standardized test, the MABC-2 test, was used to assess motor coordination in this study [17]. Despite a lack of a culturally appropriate norm, it has been validated to identify motor difficulties in Taiwanese children, showing excellent internal consistency (*α* = 0.90) and test-retest reliability (*r* = 0.88 to 0.99) and acceptable discriminative validity [18]. There were three age bands in the MABC-2 test, including 4–6 years, 7–10 years, and 11–16 years, and there were eight testing items in each age band, including three items in the subtest of manual dexterity, two in the aiming & catching, and three in balance. Children’s motor coordination was assessed using the age-appropriate testing items and categorized into either TD or rDCD based on the manual of the MABC-2 test [17]. Children were identified as rDCD if their scores were at or below the 16th percentile in this study. As this study did not attempt to make a diagnosis for DCD, this cut-off was chosen based on international clinical practice recommendations [19,20]. Preschool children with rDCD were further confirmed to have normal intelligence (i.e., IQ > 70) using the Chinese version of the Test of Nonverbal Intelligence—Fourth Edition [21]. They were also reported by their parents to have no neurological or musculoskeletal disorders that might impact their motor abilities using a medical questionnaire. Furthermore, as all school-aged children with rDCD attended mainstream elementary school, they were assumed and further confirmed by school teachers to have no physical and intellectual impairments. It is worth noting that, although the diagnostic criteria for DCD were carefully evaluated, this study used the term ‘*rDCD*’ as these children did not obtain a formal diagnosis from physicians or pediatricians.

### 2.4. Statistical Analysis

Statistical analyses were conducted on the SPSS 22.0 for Windows. Descriptive statistics were used to describe demographic information in children. Mann-Whitney U test and Chi-square statistics were used to examine the differences in demographic variables between groups. Chi-square statistics were also used to test the difference in the prevalence of DCD between preschool and school-aged children. In order to understand the effect of group (i.e., TD and rDCD), sex, and age (i.e., preschool and school-aged) on unhealthy weight status, logistic regression was separately conducted for underweight and overweight/obesity. For each dependent variable, three regression models were created. In Model 1, the group was added as the single predictor. Sex and age were further added to Model 2, while their interactions with the group (i.e., group by sex, group by age, and group by sex by age) were included in Model 3. Noteworthily, as the simultaneous inclusion of children with underweight and overweight/obesity could potentially bias the results, children with overweight/obesity were excluded while conducting logistic regression for underweight, and vice versa. The odds ratio (OR) was reported to estimate the risk for underweight and overweight/obesity, respectively.

## 3. Results

### 3.1. Overview of Participants

There were no significant differences in age, body height, body weight, BMI, and weight status between these two groups (Table 1). However, there were significantly more boys in the rDCD group (49.9% in TD vs. 71.2% in rDCD; *x*^2^ = 13.15, *df* = 1, *p* < 0.001). Furthermore, compared to TD children, children with rDCD scored significantly lower on each subtest and the overall test (all *p*-values < 0.001).

### 3.2. Prevalence of Preschool and School-Aged Children with rDCD

A total of eighty children (9.7%) were identified as rDCD (Table 1), including 56 children (6.8%) whose scores on the MABC-2 test were between the 6th and 16th percentile and 24 children (2.9%) scoring below the 6th percentile. The estimated prevalence did not significantly differ between the two age groups (*n* = 32, 10.3% in preschool children vs. *n* = 48, 9.3% in school-aged children; *x*^2^ = 0.22, *df* = 1, *p* = 0.629). However, there seemed to be more severe cases (<6th percentile) in preschool children with rDCD (*n* = 13, 40.6%) than in school-aged children with rDCD (*n* = 11, 22.9%).

### 3.3. DCD and Underweight

The overall prevalence of underweight was 12.5% (103/825, 12.3% in TD vs. 13.8% in rDCD, Table 1). The prevalence was 3.5% (11/310) in preschool children and 17.9% (92/515) in school-aged children, whereas 12.1% (52/429) of boys and 12.9% (51/396) of girls were underweight (Table 2).

As shown in Model 1 (Table 3), while children with underweight and normal weight were pooled, there was not a significant group effect, indicating that children with rDCD were not at greater risk for being underweight, compared to their TD peers. However, when sex and age were added in Model 2, the result showed that school-aged children were significantly more likely to be underweight, compared to the preschool population (*B* = 1.86, SE = 0.33, *OR* = 6.43, *p* < 0.001). While the interaction terms were further added into Model 3, a significant group effect emerged, indicating that children with rDCD were significantly more likely to be underweight than TD children (*B* = 1.89, SE = 0.75, *OR* = 6.62, *p* < 0.05), whereas age remained a significant predictor for underweight (*B* = 2.24, SE = 0.40, *OR* = 9.37, *p* < 0.001). Specifically, there was a significant interaction by age group on underweight status (*B* = −1.82, SE = 0.88, *OR* = 0.16, *p* < 0.05), indicating the potential moderating role of age. Therefore, Figure 1 was graphed to illustrate this finding, showing that while the percentage of underweight was comparable between school-aged children with and without rDCD, preschool children with rDCD were at significantly greater risk for being underweight, compared to their TD peers.

### 3.4. DCD and Overweight/Obesity

The overall prevalence of overweight/obesity was 17.0% (140/825, 16.5% in TD vs. 21.3% in rDCD, Table 1). While there were 15.2% (47/310) of preschool children and 18.1% (93/515) of school-aged children with overweight or obesity, the prevalence was 19.6% (84/429) in boys and 14.1% (56/396) in girls, respectively (Table 2).

Neither the main effect of the group nor its interactions with sex and/or age were found to significantly predict overweight/obesity (Table 4). However, Model 2 revealed a significant age effect, indicating that, regardless of motor coordination, older children were significantly at greater risk for overweight or obesity (*B* = 0.40, SE = 0.20, *OR* = 1.50, *p* < 0.05). Furthermore, there was a significant sex effect in Model 3, indicating that girls were less likely to be overweight or obese (*B* = −0.43, SE = 0.20, *OR* = 0.65, *p* < 0.05).

## 4. Discussion

This study updated the prevalence of rDCD and unhealthy weight status in a large sample of Taiwanese children. Specifically, to the best of our knowledge, this might be one of the first studies investigating underweight and overweight/obesity in children with DCD in an Asian cultural context. This study has found a lower prevalence of rDCD in both preschool and school-aged Taiwanese children in a non-representative cohort, compared to those in previous studies [22,23]. Furthermore, contrary to our hypothesis, children with rDCD are not more overweight or obese than their peers. However, younger children with rDCD might be at greater risk of being underweight.

Prior research has collected data on motor coordination in school-aged Taiwanese children between 2004 and 2006 and found that more than 20% of children were identified as DCD using the Movement Assessment Battery for Children test [22]. However, this up-to-date investigation shows that only 9.7% of children have motor difficulties. It is still being determined whether this discrepancy may be attributed to the use of different versions of the motor test or the overall improvement in motor coordination of children in the current generation. As a lack of a representative sample of preschool and school-aged children, further research with a larger sample size is warranted to affirm this prevalence, and it is needed to make a between-cohort comparison using the same assessment.

It is worth noting that while an increase in the prevalence with age was previously found based on the cut-off of the 15th percentile of the MABC testing results (5% in 4- to 6-year-olds, 13.2% in 7- to 8-year-olds, and 48.4% in 11- to 12-year-olds) [22,23,24], a similar prevalence was found for both age groups. However, the percentage of severe rDCD (i.e., <6th percentile) seems to be higher in younger children. There is a lack of research investigating the age effect on the prevalence in one single study, and this limits our ability to make a comparison with previous findings. Nevertheless, as children with DCD may not grow out of poor motor coordination, it is anticipated to see a slight change in the prevalence across ages. Unfortunately, the cross-sectional design hinders this study from tracking the over-time change in the prevalence, highlighting the need for longitudinal research. In addition, sex could play a critical role in the prevalence of DCD. Prior research has shown that DCD may be more prevalent in boys than girls [25,26]. This is also the case in this study. While the ratio of boys to girls ranges from 2:1 to 9:1 [25,26], this study has found a similar result, indicating that the boy-girl ratio is approximately 2.5:1.

The novel finding of this study is that preschool children with rDCD may be at increased risk for being underweight. When the percentage of underweight preschool children with rDCD is comparable to school-aged children with and without rDCD, it is much higher when compared to preschool children without rDCD. It is often suggested that, during early or middle childhood, children with DCD have poor performance on self-care skills, such as tying shoelaces, zipping a jacket, or buttoning clothes [26]. These impairments of fine motor skills may consequently lead to a detrimental effect on their healthy weight, as previous qualitative studies have indicated that children with DCD may be messy eaters with slowness in eating and an awkward manner of using utensils [27]. However, as the prevalence is similar in both school-aged children with and without rDCD, this interpretation may be specific only to the younger population. Notably, as other psychosocial (e.g., body image) or cultural factors could also impact weight status [28], further research may systematically take into account these effects and explore how they interact with motor coordination and determine underweight in children at early ages.

Our understanding is still limited toward childhood underweight in younger children with motor difficulties, specifically DCD. In a group of Brazilian children aged between 7 and 10 years, boys and younger children were found to be less likely to have low weight [29], whereas underweight may be less prevalent in girls in South and West Asian countries [30]. Furthermore, poor flexibility may be associated with being underweight in Italian preschool children, specifically boys [31]. The co-occurrence of other neurodevelopmental disorders, such as attention deficit hyperactive disorder, could further increase the risk for underweight in school-aged children [32]. Taking together, as these aforementioned physiological or contextual factors may also be the correlates of DCD [25,33], their moderating effects have not been well examined on the development of underweight in children with DCD. More research is definitely needed to enhance our understanding of the underlying mechanisms.

Surprisingly, inconsistent with prior cross-sectional and longitudinal studies showing that children with DCD were more likely to have higher BMI or be overweight or obese [11,14,34,35], preschool and school-aged children with DCD in this study do not tend to be overweight or obese. This finding was similar to those in two recent studies which found no significant difference in BMI in Canadian preschool and Chinese school-aged children [36,37]. We argue that this discrepancy may be due to the exclusion of underweight children while examining the difference in the rate of overweight/obesity between children with and without rDCD. By doing so, this study could more accurately examine the prevalence of overweight/obesity in both groups. However, as the prevalence of underweight and overweight/obesity may differ and be affected by country [1,6], it is strongly recommended that researchers could re-visit data and disentangle the potential impact of the inclusion of underweight children on the investigation of overweight or obesity in children with DCD, and examine whether the similar finding to this study also exists in different cultural context.

There are a few limitations that need to be addressed in this study. First, as aforementioned, there needs to be more representativeness of our sample as preschool and school-aged children were respectively recruited from two cities. This limits our ability to generalize our results to the nationwide childhood population. Future studies are needed to recruit a representative sample. Second, the underlying mechanisms of underweight and overweight/obesity are complicated and could be affected by various health factors, such as physical activity or fitness. By including and investigating such modifiable factors, it would be more helpful to provide the potential intervention implications for children with unhealthy weight, specifically those younger children with underweight. Therefore, it is recommended that further research could collect data on the determinants of unhealthy weight to better understand the underlying mechanism and guide the intervention.

## 5. Conclusions

In summary, the findings of this study highlight the increased risk of being underweight in preschool children at risk for DCD. This has important practical implications for health professionals, as it emphasizes the need for early intervention targeting these young children to prevent the development of underweight. Additionally, the study emphasizes the importance of considering both ends of the weight spectrum, including underweight and overweight/obesity, in future research on unhealthy weight status in children with DCD. By simultaneously investigating these two extreme groups, a more comprehensive understanding of weight-related issues in children with DCD can be obtained. Overall, the study contributes valuable insights for both healthcare practitioners and researchers, shedding light on the specific challenges related to weight status in children with DCD and highlighting the importance of early intervention and comprehensive research in this population.

## Figures and Tables

**Figure 1 children-10-01171-f001:**
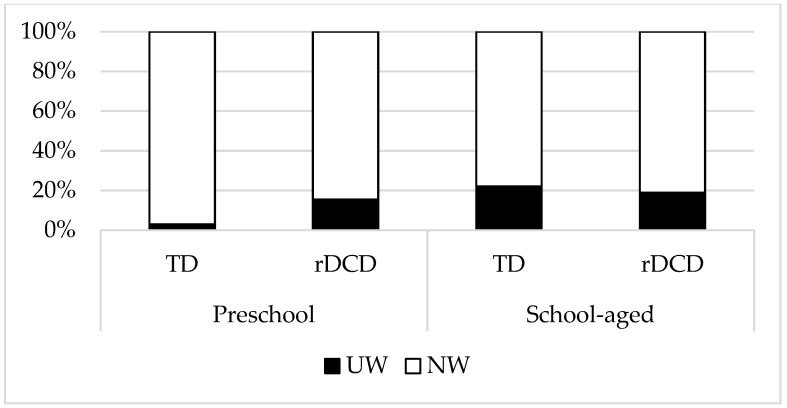
The percentage of children with normal weight and underweight. NW, normal weight; rDCD, at risk for developmental coordination disorder; TD, typical development; UW, underweight.

**Table 1 children-10-01171-t001:** Demographic information in children.

	TD (*n* = 745)	rDCD (*n* = 80)	*p* Value
Age			
Years	8.37 ± 2.69	7.97 ± 2.49	0.090
Preschool (*n*, %) ^a^	278 (37.3%)	32 (40.0%)	0.629
Boys (*n*, %) ^a^	372 (49.9%)	57 (71.2%)	<0.001
Height (cm)	128.84 ± 16.95	127.48 ± 17.52	0.591
Weight (kg)	28.68 ± 11.29	28.53 ± 11.32	0.844
Body mass index	16.65 ± 3.21	16.83 ± 3.04	0.418
Weight status ^a^			
Underweight	92 (12.3%)	11 (13.8%)	0.483
Normal weight	530 (71.1%)	52 (65.0%)	
Overweight/obesity	123 (16.5%)	17 (21.2%)	
MABC-2 component score			
Manual dexterity	31.20 ± 5.10	19.29 ± 5.07	<0.001
Aiming & catching	19.45 ± 4.88	12.94 ± 3.99	<0.001
Balance	34.40 ± 3.37	26.53 ± 6.26	<0.001
Total score	85.06 ± 8.71	58.75 ± 7.59	<0.001

^a^ Chi-square statistic was performed. rDCD, at risk for developmental coordination disorder; MABC-2, Movement Assessment Battery for Children—Second edition; TD, typical development.

**Table 2 children-10-01171-t002:** Weight status in preschool and school-aged children at different ages (boys/girls).

	Preschool		School-Aged		Total
	TD	rDCD	TD	rDCD	
underweight	6/1	3/1	37/48	6/1	52/51
normal weight	102/128	16/6	154/146	21/9	293/289
overweight/obesity	27/14	5/1	46/36	6/5	84/56
Total	135/143	24/8	237/230	33/15	429/396

rDCD, at risk for developmental coordination disorder; TD, typical development.

**Table 3 children-10-01171-t003:** Predictability of group, sex, and age on underweight.

	Model 1		Model 2		Model 3	
	*B* (SE)	*OR*	*B* (SE)	*OR*	*B* (SE)	*OR*
Group						
TD	reference		reference		reference	
rDCD	0.20 (0.35)	1.22	0.27 (0.37)	1.32	1.89 (0.75)	6.62 *
Sex						
Boy			reference		reference	
Girl			0.06 (0.22)	1.07	0.13 (0.24)	1.14
Age						
Preschool			reference		reference	
School-aged			1.86 (0.33)	6.43 ***	2.24 (0.40)	9.37 ***
Group × sex					−0.25 (0.1.27)	0.78
Group × age					−1.82 (0.88)	0.16 *
Group × sex × age					−0.83 (1.70)	0.44
Constant	−1.75 (0.11)	0.17 ***	−3.20 (0.34)	0.04 ***	−3.57 (0.41)	0.03 ***

* *p* < 0.05, *** *p* < 0.001. *B*, unstandardized coefficient; *OR*, odds ratios; rDCD, at risk for developmental coordination disorder; SE, standard error; TD, typical development.

**Table 4 children-10-01171-t004:** Predictability of group, sex, and age for overweight/obesity.

	Model 1		Model 2		Model 3	
	*B* (SE)	*OR*	*B* (SE)	*OR*	*B* (SE)	*OR*
Group						
TD	reference		reference		reference	
rDCD	0.34 (0.30)	1.41	0.28 (0.30)	1.32	0.35 (0.55)	1.42
Sex						
Boy			reference		reference	
Girl			−0.36 (0.19)	0.70 ^†^	−0.43 (0.20)	0.65 *
Age						
Preschool			reference		reference	
School-aged			0.40 (0.20)	1.50 *	0.41 (0.21)	1.51 ^†^
Group × sex					−0.20 (1.21)	0.82
Group × age					−0.50 (0.72)	0.61
Group × sex × age					1.29 (1.40)	3.65
Constant	−1.46 (0.10)	0.23 ***	−1.54 (0.19)	0.21 ***	−1.52 (0.19)	0.22 ***

^†^ *p* < 0.10; * *p* < 0.05, *** *p* < 0.001. *B*, unstandardized coefficient; *OR*, odds ratios; rDCD, at risk for developmental coordination disorder; SE, standard error; TD, typical development.

## Data Availability

The data presented in this study are available on request from the corresponding author.

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
