# Peer review of "Developmental Coordination Disorder and Unhealthy Weight Status in Taiwanese Children: The Roles of Sex and Age"

_children, 2023, doi:10.3390/children10071171_

Round 1
Reviewer 1 Report
Dear authors,
First of all, thank you for the opportunity to review this article. The present article is associates the prevalence of DCD with unhealthy weight, taking into account age and sex.
In general, the article is very well elaborated. It is a well conducted study; the aim and object of the research is understandable. The results are clearly presented and the discussion is clear. The drawbacks and limitations are also clearly stated by the authors.
In my humble opinion, this is a suitable work for the journal and is ready for publication. Just proof read: Line-116 – correct “… was choose…” – was chosen
I would like to congratulate the authors for the work. Well done!
Author Response
Dear the Reviewer,
We have provided our responses to your comments (please find the attached file). We hope that you find these adjustments satisfactory and that the revised version will be suitable for publication.
Best wishes,
The Authors

Reviewer 2 Report
Dear authors,
I have provided comments within the attached PDF. The paper is well written, however, consider the suggested changes. In addition, check all references and try to ensure that most of your references are less than 10 years old.
Kindest regards

Author Response

(The authors gave the same response as above.)

Reviewer 3 Report
The topic of the paper is interesting and fits the scope of the journal. The text is relatively well written and composed. I have only minor comments that I believe that help to improve the paper.
Introduction
Lines 33-34. Please explain better these sentences.
Materials and Methods
Lines 76-77. The inclusion of children from only two research sites may not fully represent the entire Taiwanese population.
Lines 91-92. 825 of 834 included in this study. Four of children missed body height and weight. Please explain why didn’t include the remaining 5 children?
Lines 96-97. Please replace the following sentence Children’s body height and weight were measured to calculate BMI, which was used to define weight status in this study.” With “In this study, children's body height and weight were measured to calculate their BMI (Body Mass Index).
Conclusion
Line 294. Please replace “To sum up” with “In summary”.
I think it’s better to rewrite the section of conclusion, for example like the following paragraph.
In summary, the findings of the study highlight the high risk of underweight in preschool children with rDCD (children at risk for developmental coordination disorder). This has important implications for health professionals, as it emphasizes the need for early intervention targeting these young children to prevent the development of underweight. Additionally, the study emphasizes the importance of considering both ends of the weight spectrum, including underweight and overweight/obesity, in future research on unhealthy weight status in children with DCD. By simultaneously investigating these two extreme groups, a more comprehensive understanding of weight-related issues in children with DCD can be obtained. Overall, the study contributes valuable insights for both healthcare practitioners and researchers, shedding light on the specific challenges related to weight status in children with developmental coordination disorder and highlighting the importance of early intervention and comprehensive research in this population.
The topic of the paper is interesting and fits the scope of the journal. The text is relatively well written and composed. I have only minor comments that I believe that help to improve the paper.
Introduction
Lines 33-34. Please explain better these sentences.
Materials and Methods
Lines 76-77. The inclusion of children from only two research sites may not fully represent the entire Taiwanese population.
Lines 91-92. 825 of 834 included in this study. Four of children missed body height and weight. Please explain why didn’t include the remaining 5 children?
Lines 96-97. Please replace the following sentence Children’s body height and weight were measured to calculate BMI, which was used to define weight status in this study.” With “In this study, children's body height and weight were measured to calculate their BMI (Body Mass Index).
Conclusion
Line 294. Please replace “To sum up” with “In summary”.
I think it’s better to rewrite the section of conclusion, for example like the following paragraph.
In summary, the findings of the study highlight the high risk of underweight in preschool children with rDCD (children at risk for developmental coordination disorder). This has important implications for health professionals, as it emphasizes the need for early intervention targeting these young children to prevent the development of underweight. Additionally, the study emphasizes the importance of considering both ends of the weight spectrum, including underweight and overweight/obesity, in future research on unhealthy weight status in children with DCD. By simultaneously investigating these two extreme groups, a more comprehensive understanding of weight-related issues in children with DCD can be obtained. Overall, the study contributes valuable insights for both healthcare practitioners and researchers, shedding light on the specific challenges related to weight status in children with developmental coordination disorder and highlighting the importance of early intervention and comprehensive research in this population.
Author Response

(The authors gave the same response as above.)

Round 2
Reviewer 3 Report
Thank you for this work. The manuscript can be accepted in the present form.